# FIXED STRENGTH OPTIMIZATION ENHANCES ADVERSARIAL ATTACKS

## ABSTRACT

Gradient-based multi-step iteration has been widely used to enhance attack efficiency of adversarial examples. In this work, we propose a *Fixed Strength Optimization* (FSO) method to accelerate the convergence of adversarial examples with a fixed preset attack strength. FSO can be easily combined with existing attack techniques to achieve fast convergence and well-controlled attack strength. We further introduce a combined norm based on $L_2$ and $L_\infty$ norms to modulate the attacking direction. This combined norm can help to balance the attack strength in the directions of semantic information and noise components in the model gradients. By incorporating the combined norm into FSO, our numerical experiments show improved attack transferability and high imperceptibility of perturbations.

## 1 INTRODUCTION

Adversarial examples Szegedy et al. (2013); Goodfellow et al. (2014); Bai et al. (2019) and the transferability of adversarial perturbations Goodfellow et al. (2014); Liu et al. (2016) initially raised significant security concerns Sharif et al. (2016); Eykholt et al. (2018); Ma et al. (2021) of deep neural networks (DNNs). However, following studies and applications have demonstrated various benefits derived from adversarial examples. For instance, training with adversarial examples enhances DNN robustness Madry et al. (2017); Ma et al. (2018); Wang et al. (2021b); Ilyas et al. (2018a); Duan et al. (2020); Wu et al. (2020b); Ma et al. (2021); Wang et al. (2019), allowing stable performance against unknown or malicious inputs. Recent studies also showed that adversarial examples can protect intellectual property by preventing diffusion models from generating painting imitations Liang et al. (2023). In these applications, it is essential to develop methods that generate highly transferable adversarial examples in an efficient way.

This paper considers generating adversarial examples with a constant fixed attack strength. The motivation arises from observations regarding existing attack methods during multi-step attacks. To illustrate our observations, we conducted an experiment using black-box attacks across four methods (Projected Gradient Decent (PGD) Madry et al. (2017), Skip Gradient Method (SGM) Wu et al. (2020a), Variance Reduction Method (VR) Wu et al. (2018), Interaction Reduction Method (IR) Wang et al. (2021a)). As we can see from Figure 1 (a), (1) Given an attack method, the actual perturbation strength (perturbation radius) gradually increases during the multi-step process until it reaches the preset upper limit, or never reaches the upper limit until the end; (2) For each attack method, the increase in transferability is accompanied by the increase in attack strength; (3) Methods having larger perturbation strength achieve higher transferability. These observations lead us to the question: *Is the "poor" transferability of a few methods simply due to their low perturbation strength of the adversarial examples?* For instance, the PGD method exhibits the lowest transferability, yet it also maintains the lowest perturbation strength, indicating the potential of improving its transferability by simply increasing the perturbation strength.

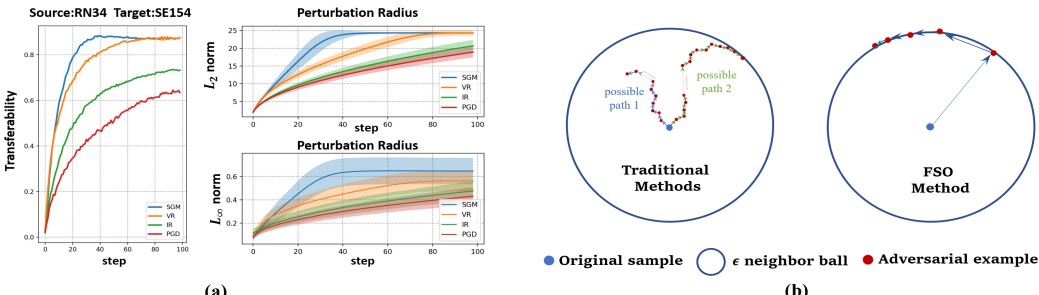

(a)

(b)

Figure 1: **(a) left:** The black-box success rate (*i.e.*, transferability) over 100 steps. [1]**(a) right:** Perturbation radius, characterized by the mean of the $L_2/L_\infty$ norm of perturbations across all samples. The shaded area represents the standard deviation. **(b):** Intuitive illustration of attack strategies. Traditional methods apply incremental perturbations from the original sample $x_{ori}$. In contrast, FSO optimizes adversarial examples on the sphere.

Motivated by this observation, we propose the *Fixed Strength Optimization* (FSO) method to directly optimize adversarial examples on the $\varepsilon$-neighboring sphere of the original sample, where the attacking strength $\varepsilon$ is the norm of the perturbation (as depicted in Figure 1 (b)). In the optimization, we simply use the tangential component of the gradient (or the variants of gradient as proposed in SGM, VR, IR, *etc.*) to update the adversarial example and use projection operation to keep it on the sphere. Compared to previous multi-step methods, FSO can achieve both faster convergence and higher transferability under the same attack strength. Furthermore, FSO allows for a fairer comparison of different attack methods since the perturbation strength is fixed.

In defining adversarial examples, a distance metric is required to quantify similarity. A common choice of the metric would be $L_p$ norm. Here we point out a few facts about the $L_2$ and $L_\infty$ norms, which are the most used metric: (1) The optimization in FSO is difficult if the $L_\infty$ norm is used, because the tangential component is always zero; (2) A perturbation obtained with a multi-step $L_\infty$ attack method is not an optimized perturbation based on $L_\infty$ norm. Namely, the perturbation on many pixels are not maximized because the sign of the perturbation on these pixels changes in different steps; (3) Adversarial examples obtained by $L_2$ attack method, while achieve higher transferability compared to those obtained by $L_\infty$ attack method, also maintain greater visual impairment. As a general assumption of gradient-based post-hoc interpretability methods Simonyan et al. (2013); Smilkov et al. (2017), the magnitude of the input-gradient highlights task-relevant features. Coordinates with larger input-gradient magnitude contain more semantic information, thus being more relevant to model predictions than those with smaller magnitude.

In this work, we propose a combined norm, the $L_{2\_\infty}$ norm, to quantify the strength of perturbations. The combined norm is determined by both the $L_2$ norm and $L_\infty$ norm. As a consequence, the combined norm helps to inhibit perturbations on large input-gradient directions compared to attacks based on $L_2$ norm. This is helpful to suppress attacks in the direction of semantic information. The new norm also helps to inhibit strong perturbations on the small input-gradient directions compared to attacks based on $L_\infty$ norm. This prevents from introducing strong noise into adversarial examples. As a side benefit, the new norm can be smoothly incorporated in FSO.

Our contributions are summarized as follows.

- We introduce FSO to generate adversarial examples in an optimization fashion under fixed attacking strength. FSO generates adversarial examples in only a few iteration steps, whereas improves the state-of-the-art transferability benchmarks.

- We introduce combined norm that suppresses the shortcomings of $L_2$ and $L_\infty$ norms in generating adversarial examples. By incorporating the combined norm into FSO, perturbations in our numerical experiments demonstrate enhanced attack transferability and high imperceptibility.

---

[1]The attacks are crafted on 1000 ImageNet validation images Wang et al. (2021a) under maximum $L_2$ perturbation $\varepsilon = 16$ (pixel values range from [0, 255]), corresponding to $L_2$ radius $r = \frac{\varepsilon}{255} \cdot \sqrt{\dim} \approx 24.3438$. All four methods are executed under the same parameter settings (perturbation radius, step size, number of iterations, etc.). The black-box success rate is tested against a 154 layer Squeeze-and-Excitation network (SE154) Hu et al. (2018), using an ImageNet-trained ResNet-34 as the source model.

## 2 RELATED WORKS

### 2.1 MULTI-STEP ATTACKS AND ADVERSARIAL TRANSFERABILITY.

Given a clean example $\boldsymbol{x}_{ori}$ with class label $y$ and a target DNN model $f$, the goal of an adversary is to find an adversarial example $\boldsymbol{x}_{adv}$ that fools the network into making an incorrect prediction (*i.e.* $f(\boldsymbol{x}_{adv}) \neq y$), while still remaining in the $\varepsilon$-ball centered at $\boldsymbol{x}_{ori}$ (*i.e.* $\|\boldsymbol{x}_{adv} - \boldsymbol{x}_{ori}\| \leq \varepsilon$). Existing adversarial attacks can be broadly categorized into two types: white-box attacks Goodfellow et al. (2014); Madry et al. (2017); Kurakin et al. (2018); Papernot et al. (2016); Su et al. (2019); Carlini & Wagner (2017); Szegedy et al. (2016); Chen et al. (2018); Modas et al. (2019), where the adversary has full access to the target model, and black-box attacks Liu et al. (2016); Ilyas et al. (2018a); Chen et al. (2017a); Bhagoji et al. (2018); Papernot et al. (2017); Bai et al. (2020), where the adversary has no information about the target model. A specific type of black-box attack leverages adversarial transferability Wu et al. (2020a; 2018); Xie et al. (2019); Dong et al. (2018), where adversarial perturbations generated on a source DNN are transferred to other target DNNs.

**Fast Gradient Sign Method (FGSM)** Goodfellow et al. (2014). FGSM lays the groundwork for gradient-based adversarial attacks by perturbing clean example $x_{ori}$ by the amount of $\varepsilon$ along the gradient direction:

$$\boldsymbol{x}_{adv} = \boldsymbol{x}_{ori} + \boldsymbol{\delta}, \boldsymbol{\delta} = \varepsilon \cdot \operatorname{sign}\left(\nabla_{\boldsymbol{x}} \ell\left(f\left(\boldsymbol{x}_{ori}\right), y\right)\right),$$

where $\ell$ is objective function. The Basic Iterative Method (BIM) Kurakin et al. (2018) is an iterative version of FGSM that perturbs for $T$ steps with step size $\varepsilon/T$.

Based on these insights, researchers have developed multi-step attack methods Madry et al. (2017); Kurakin et al. (2018); Carlini & Wagner (2017); Chen et al. (2018) to generate more potent adversarial examples. These multi-step approaches typically perturb normal example $x_{ori}$ for $T$ steps with smaller step size $\alpha$ (different to BIM, $\alpha > \varepsilon/T$ is allowed). After each step of perturbation, if the adversarial example goes out of the $\varepsilon$-ball of $x_{ori}$, it is projected back to the $\varepsilon$-sphere.

**Multi-step gradient-based attack methods have a generalized formula structured as follows:**

$$\boldsymbol{x}_{adv}^{t+1} = \Pi_{\varepsilon}\left(\boldsymbol{x}_{adv}^{t} + \boldsymbol{\delta}^{t}\right), \boldsymbol{\delta}^{t} = \alpha \cdot \mathcal{N}_{p}\left(\boldsymbol{g}^{t}\right), \tag{1}$$

where $\Pi_{\varepsilon}(\cdot)$ is the projection operation, $\boldsymbol{g}^{t}$ is the perturbation direction, $\mathcal{N}_{p}(\cdot)$ is a normalization operator tailored for various $p$-norms, adjusting both the direction and magnitude of $\boldsymbol{g}^{t}$ to meet the constraints imposed by each norm. For example, $\mathcal{N}_{\infty}(\boldsymbol{g}^{t}) = \operatorname{sign}(\boldsymbol{g}^{t})$ limits each element's magnitude, $\mathcal{N}_{2}(\boldsymbol{g}^{t}) = \frac{\boldsymbol{g}^{t}}{\|\boldsymbol{g}^{t}\|_{2}}$ ensures uniform scaling, and $\mathcal{N}_{1}(\boldsymbol{g}^{t})$ selectively modifies the largest components to enforce sparsity.

The essence of multi-step attack methods lies in the design of the perturbation direction $\boldsymbol{g}^{t}$ to iteratively refine adversarial perturbations, thereby enhancing their transferability and increasing the difficulty for models to defend against them. Notable examples include:

**Projected Gradient Descent (PGD)** Madry et al. (2017). PGD directly uses the gradient as a perturbation direction:

$$\boldsymbol{g}^{t} = \nabla_{\boldsymbol{x}} \ell\left(f\left(\boldsymbol{x}_{adv}^{t}\right), y\right).$$

**Momentum Iterative boosting (MI)** Dong et al. (2018). MI incorporates a momentum term into the gradient to stabilize update directions, thereby boosting the transferability:

$$\boldsymbol{g}^{t} = \mu \cdot \boldsymbol{g}^{t-1} + \frac{\nabla_{\boldsymbol{x}} \ell\left(f\left(\boldsymbol{x}_{adv}^{t}\right), y\right)}{\left\|\nabla_{\boldsymbol{x}} \ell\left(f\left(\boldsymbol{x}_{adv}^{t}\right), y\right)\right\|_{1}},$$

where $\boldsymbol{g}^{t-1}$ represents the perturbation direction from the previous step, $\mu$ is the decay factor, and $\|\cdot\|_{1}$ denotes the $L_{1}$ norm.

**Diverse Input (DI)** Xie et al. (2019). DI proposes to craft more universally effective perturbations using gradient with respect to the randomly-transformed input example:

$$\boldsymbol{g}^t = \nabla_{\boldsymbol{x}} \ell \left( f \left( H \left( \boldsymbol{x}_{adv}^t; p \right) \right), y \right),$$

where $H \left( \boldsymbol{x}_{adv}^t; p \right)$ is a stochastic transformation function on $\boldsymbol{x}_{adv}^t$ for a given probability $p$.

**Translation Invariant (TI)** Dong et al. (2019). TI targets to evade robustly trained DNNs by generating adversarial examples that are less sensitive to the discriminative regions of the surrogate model. More specifically, TI computes the gradients with respect to a set of translated versions of the original input:

$$\boldsymbol{g}^t = \boldsymbol{W} * \nabla_{\boldsymbol{x}} \ell \left( f \left( \boldsymbol{x}_{adv}^t \right), y \right),$$

where $\boldsymbol{W}$ is a predefined kernel (*e.g.*, uniform, linear, and Gaussian) matrix of size $(2k+1)(2k+1)$ ($k$ being the maximal number of pixels to shift). This kernel convolution is equivalent to the weighted sum of gradients over $(2k+1)^2$ shifted input examples.

**Variance Reduction (VR)** Wu et al. (2018). VR employs smoothed gradients to generate perturbations with high transferability by smoothing the classification loss with Gaussian noise during attacking:

$$\boldsymbol{g}^t = \mathbb{E}_{\boldsymbol{\xi} \sim N(\boldsymbol{0}, \sigma^2 \boldsymbol{I})} \left[ \nabla_{\boldsymbol{x}} \ell \left( f \left( \boldsymbol{x}_{adv}^t + \boldsymbol{\xi} \right), y \right) \right],$$

where $\mathbb{E}_{\boldsymbol{\xi} \sim N(\boldsymbol{0}, \sigma^2 \boldsymbol{I})}$, indicates that an expectation (or average) is taken over Gaussian noise $\boldsymbol{\xi}$, sampled from a multivariate normal distribution $N(\boldsymbol{\mu} = \boldsymbol{0}, \sigma^2 \boldsymbol{I})$. The noise $\boldsymbol{\xi}$ is added to the adversarial example to smooth the gradient.

**Skip Gradient Method (SGM)** Wu et al. (2020a). SGM proposes to enhance the transferability of adversarial examples by using the gradients of skip connections more than those of residual modules:

$$\boldsymbol{g}^t = \nabla_{\boldsymbol{x}}^{\text{skip}} \ell \left( f \left( \boldsymbol{x}_{adv}^t \right), y \right),$$

where $\nabla_{\boldsymbol{x}}^{\text{skip}} \ell = \frac{\partial \ell}{\partial \boldsymbol{z}_L} \prod_{i=0}^{L-1} \left( \gamma \frac{\partial f_{i+1}}{\partial \boldsymbol{z}_i} + 1 \right) \frac{\partial \boldsymbol{z}_0}{\partial \boldsymbol{x}}$ represents the neural network function that prioritizes gradients from skip connections over those from residual connections in $L$ residual blocks. Here, $\boldsymbol{z}_0 = \boldsymbol{x}$ is the network input, and $\gamma \in (0, 1]$ is the decay parameter to reduce the gradient from the residual modules.

Furthermore, other adversarial attach methods include 1) sparsity-based methods such as Jacobian-based Saliency Map Attack (JSMA) Papernot et al. (2016), sparse attack Modas et al. (2019), one-pixel attack Su et al. (2019), 2) optimization-based methods such as Carlini and Wagner (CW) Carlini & Wagner (2017) and elastic-net (EAD) Chen et al. (2018), decoupled direction and norm (DDN) attack Rony et al. (2019), 3) query-based methods Chen et al. (2017a); Ilyas et al. (2018b; 2019); Uesato et al. (2018); Andriushchenko et al. (2020), 4) gradient estimation methods such as Finite Differences(FD) Chen et al. (2017a); Bhagoji et al. (2018) or Natural Evolution Strategies (NES) Ilyas et al. (2018a); Jiang et al. (2019), and 5) intermediate features-based methods such as Activation Attack Inkawhich et al. (2019) and Intermediate Level Attack Huang et al. (2019). In particular, Interaction Reduction (IR) Attack Wang et al. (2021a) reduces interactions between input units to create more transferable adversarial examples.

## 3 FIXED STRENGTH OPTIMIZATION

### 3.1 MOTIVATION

The generation of adversarial examples can be obtained by the following optimization equation:

$$\boldsymbol{\delta} = \arg \max_{\|\boldsymbol{\delta}\| \leq \varepsilon} \ell(f(\boldsymbol{x}_{ori} + \boldsymbol{\delta}), y). \tag{2}$$

The loss function $\ell(f(\boldsymbol{x}_{adv}), y)$ measures the performance of model $f$, where $\boldsymbol{x}_{adv} = \boldsymbol{x}_{ori} + \boldsymbol{\delta}$ is the adversarial example, and $\boldsymbol{\delta}$ is the adversarial perturbation. The adversarial example lies within an $\varepsilon$-ball around $\boldsymbol{x}_{ori}$ (*i.e.*, $\|\boldsymbol{x}_{adv} - \boldsymbol{x}_{ori}\| \leq \varepsilon$).

Multi-step gradient-based attack methods are not efficient in performing the optimization in Equation (2). First, as shown in Figure 1 (a), although the optimized adversarial example is usually on the $\varepsilon$-sphere, multi-step methods requires many iteration steps to progressively increase the perturbation strength[2]; second, the tangential component (to the $\varepsilon$-sphere) of the perturbation direction $\boldsymbol{g}^t$ is usually small compared to the normal component, which lead to slow convergence of the multi-step methods. In particular, when the $L_\infty$ norm is used, adversarial examples obtained by multi-step methods usually do not stay on the $\varepsilon$-sphere, since the sign of the perturbation direction $\boldsymbol{g}^t$ on many pixels changes from step to step.

Next, we propose a method, Fixed Strength Optimization (FSO), to directly optimize adversarial examples on the $\varepsilon$-sphere, *i.e.*,$\|\boldsymbol{x}_{adv} - \boldsymbol{x}_{ori}\| \equiv \varepsilon$. Existing attack methods can be naturally incorporated to determine the tangential perturbation direction to update adversarial examples. FSO significantly reduces the number of iterations while improving the transferability.

### 3.2 ALGORITHM

In FSO, we use the tangential component of the perturbation direction $\boldsymbol{g}^t$ to update the adversarial example, where $\boldsymbol{g}^t$ can be obtained with the attack method list in Section 2. The initial example is directly obtained by scaling $\boldsymbol{g}^0$ to meet the perturbation strength. In order to keep the adversarial example on the sphere, a normalization is used for the updated example. Details of FSO is included in Algorithm 1.

---

**Algorithm 1** FSO Method for Adversarial Example Generation.

$\boldsymbol{g^t}$      Perturbation direction at step $t$.
$\alpha^t$      Step size at step $t$.
$\mathcal{N}_p(\cdot)$      Normalization operator for a general norm.
$\Pi_\epsilon(\cdot)$      Projection operator to constrain the adversarial example on the $\epsilon$-sphere.

1: **Input:** $\boldsymbol{x}_{ori}$: original sample,   $\varepsilon$: maximum perturbation constraint,   $T$: total number of attack steps,
2: Initialize: $\boldsymbol{x}_{adv}^0 \leftarrow \boldsymbol{x}_{ori}$,
3: **for** $t \leftarrow 0$ to $T-1$ **do**
4:      **if** $t = 0$ **then**
5:          $\boldsymbol{x}_{adv}^{t+1} \leftarrow \boldsymbol{x}_{adv}^t + \varepsilon \cdot \mathcal{N}_p(\boldsymbol{g}^t)$,                     ▷ Initialize to fixed strength perturbation
6:      **else**
7:          **g_normal**, **g_tangent** $\leftarrow \boldsymbol{g}^t$,                    ▷ Gradient decomposition
8:          $\boldsymbol{x}_{adv}^{t+1} \leftarrow \boldsymbol{x}_{adv}^t + \alpha^t \cdot \mathcal{N}_p(\textbf{g\_tangent})$,       ▷ Update along the tangential direction
9:          $\boldsymbol{x}_{adv}^{t+1} = \Pi_\varepsilon\left(\boldsymbol{x}_{adv}^{t+1}\right)$,                       ▷ Project to the $\varepsilon$-sphere
10:      **end if**
11: **end for**
12: **Output:** $\boldsymbol{x}_{adv}^T$ : the final adversarial example.

---

In practical implementation, for a general norm, the normal direction at $\boldsymbol{\delta}^t = \boldsymbol{x}_{adv}^t - \boldsymbol{x}_{ori}$ is simply the gradient direction $\boldsymbol{n}^t = \frac{\nabla\|\boldsymbol{\delta}^t\|}{\|\nabla\|\boldsymbol{\delta}^t\|\|}$. In particular, for the $L_2$-norm case, $\boldsymbol{n}^t = \frac{\boldsymbol{\delta}^t}{\|\boldsymbol{\delta}^t\|_2}$. The tangential component of $\boldsymbol{g}^t$ can be easily obtained by subtracting the normal component from $\boldsymbol{g}^t$. Since a normalization of the tangent component is used, we set a decaying step size $\alpha^t = \alpha_0/t$ to ensure convergence of the adversarial example.

## 4   $L_{2\_\infty}$ NORM

It is worth noting that FSO cannot be used under the $L_\infty$ norm because the tangential component is always zero. Since limiting the $L_\infty$ norm of the perturbation of adversarial examples is important in suppressing the attack of semantic information, we need further effort to deal with this case.

Meanwhile, as mentioned above and can be seen from Figure 2 (a), adversarial examples obtained by the multi-step attack methods using $L_\infty$ norm is not optimized on the corresponding $\varepsilon$-sphere (the attack strength on many pixels is smaller than the preset strength $\varepsilon_\infty = \frac{16}{255}$). This is because

---

[2]Figure 1 (a) shows that the perturbation radius of the SGM method reaches the surface of the neighboring sphere at around 40 steps, while the PGD method does not reach it even after 100 steps.

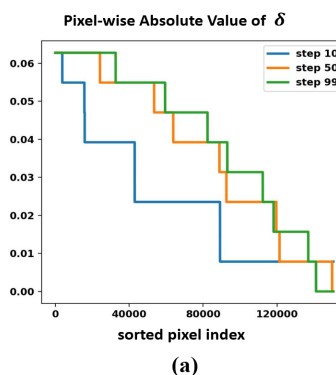
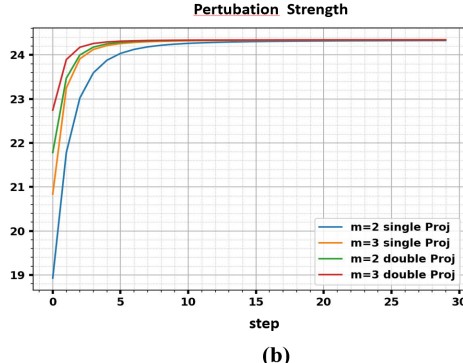

**(a)**                                              **(b)**

Figure 2: **(a):** The sorted pixel-wise perturbation strength obtained with multi-step PGD $L_\infty$ attack for a $d = 3 \times 224 \times 224$ image. **(b):** Convergence of the perturbation strength (the $L_{2\_\infty}$ norm) towards the preset value, $\varepsilon_\infty \cdot \sqrt{d}$.

the sign of the perturbation direction $g^t$ can change on many pixels during the multi-step process. These pixels with unstable sign of $g^t$ usually also maintains relatively small abstract value. From this point of view, the multi-step attack method based on $L_\infty$ norm indirectly suppresses the attack on the pixels with weak semantic information in fact. This suppression is helpful to prevent from introducing too strong noise (in the sense of $L_2$ norm) into the adversarial examples.

Based on the above observation, we introduce the $L_{2\_\infty}$ norm,

$$\|\boldsymbol{\delta}\|_{2\_\infty,m} = \max\{\|\boldsymbol{\delta}\|_2, \frac{\sqrt{d}}{m}\|\boldsymbol{\delta}\|_\infty\}, \tag{3}$$

where $1 \le m \le \sqrt{d}$ is a hyper-parameter, where $d$ is the dimension of $\delta$. In particular, when $m = 1$, the norm reduces to $\sqrt{d}$ times the $L_\infty$ norm; whereas when $m = \sqrt{d}$, the norm is equivalent to the $L_2$ norm. In other words, the $\varepsilon-$sphere under the $L_{2\_\infty}$ norm is obtained by cutting off the corresponding Euclidean sphere by parallel planes in each coordinate axis direction. The distance from the origin to the planes is equal to $\frac{m\varepsilon}{\sqrt{d}}$.

Therefore, compared to a perturbation $\boldsymbol{\delta}'$ obtained with the $L_2$ norm satisfying $\|\boldsymbol{\delta}'\|_2 = \|\boldsymbol{\delta}\|_{2\_\infty,m}$, $\boldsymbol{\delta}$ has smaller $L_\infty$ norm, namely, $\|\boldsymbol{\delta}\|_\infty \le \|\boldsymbol{\delta}'\|_\infty$; whereas compared to a perturbation $\boldsymbol{\delta}'$ obtained with the $L_\infty$ norm satisfying $\|\boldsymbol{\delta}'\|_\infty = \|\boldsymbol{\delta}\|_{2\_\infty,m}$, $\boldsymbol{\delta}$ has smaller $L_2$ norm, namely, $\|\boldsymbol{\delta}\|_2 \le \|\boldsymbol{\delta}'\|_2$. In particular, when we have $\|\boldsymbol{\delta}\|_{2\_\infty,m} = \varepsilon_\infty\sqrt{d}$, the largest possible $L_\infty$ norm of $\boldsymbol{\delta}$ is $m\varepsilon_\infty$.

This new norm is potential to achieve good balance of the perturbation: It both suppresses the attack on the large component directions of $\boldsymbol{\delta}$, thus helping to keep the semantic information of data, and suppresses attack on small component directions, thus reducing noise introduction on these directions. In other words, for generation of adversarial examples, the new norm achieves both advantages of $L_2$ norm and $L_\infty$ norm.

Using the $L_{2\_\infty}$ norm, the projection operator $\Pi_\varepsilon(\cdot)$ in Algorithm 1 is realized by an $L_2$ projection step ($\boldsymbol{\delta} \leftarrow \varepsilon \cdot \frac{\boldsymbol{\delta}}{\|\boldsymbol{\delta}\|_2}$) followed by an $L_\infty$ projection step (resetting the components satisfying $|\boldsymbol{\delta}_i| > \frac{m\varepsilon}{\sqrt{d}}$ to $\text{sign}(\boldsymbol{\delta}_i) \cdot \frac{m\varepsilon}{\sqrt{d}}$, where $i$ is the index of the components of $\boldsymbol{\delta}$). This realization is not the exact projection $\Pi_\varepsilon$. In fact, the $L_{2\_\infty}$ norm of the perturbation is usually smaller than $\varepsilon$ after this projection. Nevertheless, as shown in Figure 2(b), the norm converges to $\varepsilon$ quickly in the optimization process (see curves with legend "single Proj"). When $m$ is small, we can repeat the projection step for once to even accelerate the convergence (see curves with legend "double Proj").

## 5 EXPERIMENTS

Next, we use numerical experiments to show the fast convergence rate, the enhanced transferability, and the ballaced perturbation of FSO under the $L_{2\_\infty}$ norm. Unless otherwise specified, the preset attack strength under the $L_\infty$, $L_2$, and $L_{2\_\infty}$ norms are set to $\varepsilon_\infty = \frac{16}{255}$, $\varepsilon_2 = \sqrt{d}\varepsilon_\infty$, and

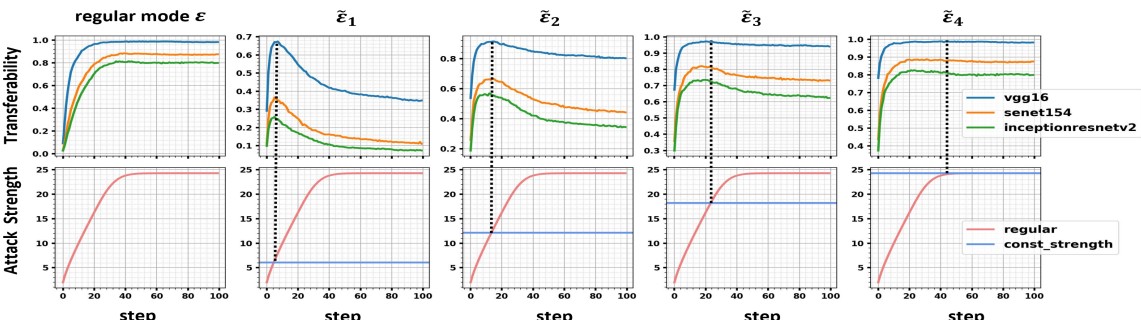

Figure 3: **Top:** The black-box transferability (*i.e.*, success rate of the transfer attack) of the adversarial examples obtained with the regular model to three target models. The left panel is for the original perturbation and the right 4 panels are for the rescaled perturbations to $\tilde{\varepsilon}_k$, $k = 1, 2, 3, 4$, respectively. The blue, orange, and green curves correspond to the transferability on VGG16, SE-154, and IncResV2, respectively. **Bottom:** Average perturbation radius $\|\boldsymbol{\delta}^t\|_2$, with the shaded area indicating the standard deviation. The horizontal blue straight lines shows the attack strength after rescaling, whereas the vertical dash-black line shows the step at which $\|\boldsymbol{\delta}^t\|_2 = \tilde{\varepsilon}_k$.

$\varepsilon_{2\_\infty} = \sqrt{d}\varepsilon_\infty$, respectively, following the setting in Dong et al. (2018). The dimension for images used in this work is $d = 3 \times 224 \times 224$. The step size in multi-step attack methods is set to 2/255.

## 5.1 Transferabiliy analysis of Multi-step Methods

We use a numerical experiment to illustrate the transferability of traditional multi-step attack methods. To this end, we first performed an $L_2$ attack using SGM Wu et al. (2020a) on a source DNN (RN-34) and transferred the adversarial perturbations to three target DNNs (VGG16 Simonyan & Zisserman (2015), SE-154 Hu et al. (2018), and IncResV2 Szegedy et al. (2017)), referred to as "regular mode". The attacks were executed over 100 steps on validation images Wang et al. (2021a) from the ImageNet dataset Russakovsky et al. (2015). To our experience, other multi-step attack methods and model architectures similar results.

The transferability to the three target models is collected in the top-left panel in Figure 3, whereas the evolution of attack strength ($\|\boldsymbol{\delta}^t\|_2$ obtained with the regular model) is shown in the bottom-left panel. We can see that the transferability increases with the step, acompanied with the increase of the attack strength.

In order to see whether the increase of the transferability is simply due to the increase of attack strength, we rescales the perturbation to different attack strengths $\tilde{\varepsilon}_k = \frac{k\varepsilon}{4}$ and $\boldsymbol{x}_{adv,k}^t = \boldsymbol{x}_{ori} + \tilde{\varepsilon}_k \cdot \frac{\boldsymbol{\delta}^t}{\|\boldsymbol{\delta}^t\|_2}$. The transferability for $k = 1, 2, 3, 4$ is included in the top-right 4 columns in Figure 3. Roughly speaking, as the iteration step increases, all transferability curves increase first to their maximums and then decrease. The maximums are obtained when $\|\boldsymbol{\delta}^t\|_2 \sim \tilde{\varepsilon}_k$.

This result shows that the optimal perturbation direction is dependent on the perturbation strength. Aside from the perturbation strength, the perturbation direction also plays an important role in the transferability. Interestingly, the transferability curves for all three models peak at almost the same step, indicating that the optimal perturbation direction is independent of the model.

Based on the above observations, we can conclude that optimizing the perturbation directly on the fixed-strength $\varepsilon-$ sphere is advantageous. It avoid the slow growth of attack strength and optimize the perturbation direction under a specific attack strength.

## 5.2 Effects of $m$ in the $L_{2\_\infty}$ norm

Based on the same source model and experiment setting as Figure 1, we conduct black-box attack to determine suitable values of $m$ introduced in the $L_{2\_\infty}$ norm. We tested the transferability of the generated perturbations on seven target DNNs, including VGG-16, ResNet-152 (RN-152), DenseNet-201 (DN-201), SENet-154 (SE- 154) Hu et al. (2018), InceptionV3 (IncV3) Szegedy et al. (2016),

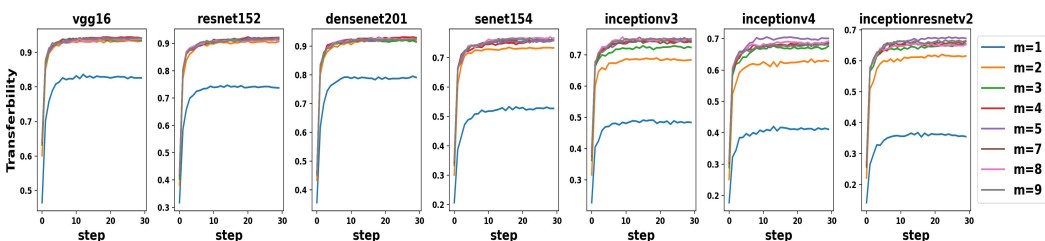

Figure 4: The transferability of FSO combined with SGM attack method, under the $L_{2\_\infty}$ norm with different $m$.

InceptionV4 (IncV4) Szegedy et al. (2017), and Inception-ResNetV2 (IncResV2) Szegedy et al. (2017).

In Figure 4, we show the transfer attack performance of FSO using the $L_{2\_\infty}$ norm with different $m$. The performance on all seven target models exhibit the same trend:

1. Each curve increases with the iteration step and converge to a plateau in a few ($\leq 10$) steps. This shows the fast convergence rate of FSO. As a comparison, the multi-step attack shown in Figure 3 requires more than 20 iteration steps to converge to the plateau. Thus FSO can accelerate the convergence (by $2 \sim 3$ folds) and reduce iteration steps. Compared to results shown in Figure 1 (a), where the attack strength may increases slowly, the acceleration rate can be even larger.

2. The transferability increases with $m$ for all test networks. For the first three target networks, $m = 2$ is sufficiently large that further increment of $m$ only leads to insignificant increase of the transferability. Since larger $m$ means larger allowed $L_\infty$ norm of the perturbation while the largest possible $L_2$ norm remains unchanged, the monotonically increasing property is natural. It is important to see that $m \sim 2$ or 3 is sufficient to achieve nearly best performance. This means that we can still control the $L_\infty$ norm of the perturbation well ($\leq m\varepsilon_\infty$), thus suppressing strong attack of semantic information. The small saturation number of $m$ allows us to achieve balanced attack–with both weak semantic attack and weak noise introduction.

In summary, we can achieve fast convergence and balanced attack using FSO under the $L_{2\_\infty}$ norm.

### 5.3   TRANSFERABILITY ENHANCEMENT OF FSO UNDER $L_{2\_\infty}$ NORM

Next we systematically study the transferability enhancement of FS Ounder $L_{2\_\infty}$ norm compared with multi-step attack methods with $L_2$ and $L_\infty$ norms. We generated adversarial perturbations on six source DNNs, including VGG16 Simonyan & Zisserman (2015), and IncResV2 Szegedy et al. (2017)), Alexnet Krizhevsky et al. (2012), ResNet-34/152 (RN-34/152) He et al. (2016), and DenseNet-121/201 (DN-121/201) Huang et al. (2017). The target test networks are the same as that in Figure 4. In additon, we used the Dual-Path-Network (DPN-68) Chen et al. (2017b) to evaluate the ensemble source model[3] following the setting in Wang et al. (2021a).

We used the widely used PGD Attack Madry et al. (2017) as the baseline method to compare the results. All attacks executed over 30 steps on validation images Wang et al. (2021a) from the ImageNet dataset Russakovsky et al. (2015). To enable fair comparisons, the transferability is computed with the best adversarial perturbation during the 30 steps via the leave-one-out (LOO) validation in Wang et al. (2021a). All attacks were conducted with three different random samplings of grids or different initial perturbations.

Table 1 reports the success rates of the multi-step PGD attack under $L_\infty$ and $L_2$ norms and FSO PGD attack under $L_2$ and $L_{2\_\infty}$ norms. Compared with the baseline attack, the transferability is significantly improved by using FSO. In particular, although constraints on the $L_\infty$ norm are applied to the perturbation in the $L_{2\_\infty}$ norm, the transferability obtained with the $L_{2\_\infty}$ norm is comparable with that obtained simply with the $L_2$ norm in FSO. Compared to the multi-step method, the large improvement is partly due to the slow increase of the perturbation strength for PGD attack method

---

[3]Besides above adversarial transferring from a single-source model, the ensemble source model Liu et al. (2016) generate adversarial perturbations on the ensemble of RN-34, RN-152, and DN-121.

Table 1: The success rates of adversarial examples generated from six source models against seven target models using PGD black-box attack under $L_\infty$ and $L_2$ norms and FSO under $L_{2\_\infty}$ norm.

| Source | Method | Target Models | | | | | | |
|---|---|---|---|---|---|---|---|---|
| | | VGG-16 | RN152 | DN-201 | SE-154 | IncV3 | IncV4 | IncResV2 |
| AlexNet | PGD $L_\infty$ | 58.9±1.2 | 22.8±0.7 | 27.2±0.9 | 23.6±0.5 | 23.2±0.4 | 19.3±0.3 | 14.7±0.5 |
| | PGD $L_2$ | 74.6±1.5 | 44.7±2.2 | 45.5±0.4 | 38.6±0.7 | 43.1±1.9 | 34.5±1.4 | 30.5±2.3 |
| | PGD+FSO $L_{2\_\infty}^{3X}$ | **87.9±1.0** | **57.7±1.5** | **59.4±0.4** | **52.5±0.6** | **58.6±0.5** | **47.3±1.5** | **42.5±1.0** |
| VGG-16 | PGD $L_\infty$ | - | 38.8±1.1 | 42.4±0.9 | 46.4±0.3 | 33.0±1.2 | 43.8±0.7 | 27.8±1.3 |
| | PGD $L_2$ | - | 52.1±1.7 | 57.6±0.4 | 56.8±0.7 | 52.4±1.5 | 59.8±0.4 | 42.7±1.4 |
| | PGD+FSO $L_{2\_\infty}^{3X}$ | - | **73.7±1.5** | **79.8±0.3** | **81.1±0.6** | **76.1±0.5** | **81.5±1.5** | **64.7±1.0** |
| RN-34 | PGD $L_\infty$ | 61.0±0.6 | 59.3±1.3 | 62.3±0.4 | 32.4±0.4 | 26.3±0.6 | 24.4±0.4 | 21.3±1.1 |
| | PGD $L_2$ | 68.6±1.2 | 70.5±1.1 | 71.4±0.8 | 39.8±0.7 | 37.8±1.5 | 36.5±2.4 | 28.1±1.5 |
| | PGD+FSO $L_{2\_\infty}^{3X}$ | **95.1±1.0** | **92.5±1.5** | **93.5±1.4** | **79.6±0.6** | **76.4±0.9** | **69.9±1.5** | **68.6±1.0** |
| RN-152 | PGD $L_\infty$ | 47.8±2.5 | - | 61.4±0.9 | 36.8±1.0 | 27.9±1.9 | 24.3±1.1 | 23.4±1.6 |
| | PGD $L_2$ | 55.6±2.6 | - | 72.1±1.2 | 45.6±1.9 | 38.9±2.2 | 37.2±1.3 | 34.1±2.4 |
| | PGD+FSO $L_{2\_\infty}^{3X}$ | **87.6±1.0** | - | **93.2±1.1** | **77.4±1.6** | **73.7±1.4** | **66.5±0.8** | **66.6 ±1.2** |
| DN-121 | PGD $L_\infty$ | 64.9±1.9 | 62.6 ±1.4 | 85.5±1.1 | 43.1±1.1 | 34.7±2.3 | 34.9±0.9 | 27.9±1.6 |
| | PGD $L_2$ | 70.4±2.6 | 70.9±1.7 | 90.2±0.9 | 48.5±1.7 | 47.4±1.4 | 45.5±1.4 | 39.2±2.3 |
| | PGD+FSO $L_{2\_\infty}^{3X}$ | **94.3±1.1** | **91.8±1.1** | **98.7±1.1** | **84.9±1.3** | **80.4±1.5** | **76.6±0.9** | **72.7 ±1.3** |
| DN-201 | PGD $L_\infty$ | 58.6±1.5 | 68.3 ±1.0 | - | 49.6±1.3 | 39.5±2.6 | 38.3±1.4 | 32.3±1.2 |
| | PGD $L_2$ | 63.7±1.9 | 76.2±1.1 | - | 55.2±2.2 | 51.0±2.4 | 49.1±1.9 | 42.9±1.6 |
| | PGD+FSO $L_{2\_\infty}^{3X}$ | **92.6±1.0** | **94.3±1.4** | - | **87.6±1.8** | **82.3±1.4** | **80.2±0.9** | **76.6 ±1.5** |

Table 2: The success rates of comparison of black-box attacks crafted on the ensemble model (RN-34+RN-152+DN-121) against target models.

| Source | Method | Target Models | | | | | | | |
|---|---|---|---|---|---|---|---|---|---|
| | | VGG-16 | RN152 | DN-201 | SE-154 | IncV3 | IncV4 | IncResV2 | DPN-68 |
| Ensemble | PGD $L_\infty$ | 85.5±1.6 | 100.0±0.2 | 97.1 ± 0.6 | 74.9±0.8 | 65.8±0.7 | 63.0±0.9 | 56.4±0.5 | 64.8±0.6 |
| | PGD $L_2$ | 87.9±0.5 | 99.6±0.2 | 98.2±0.4 | 79.8±0.7 | 75.9±0.9 | 72.8±0.4 | 68.8±0.3 | 78.2±0.5 |
| | PGD+FSO $L_2$ | **98.6±0.1** | **100±0.0** | **100.0±0.0** | **94.4±0.1** | **94.0±0.1** | **92.1±0.2** | **92.4±0.1** | **94.4±0.1** |
| | PGD+FSO $L_{2\_\infty}^{3X}$ | **98.3±0.1** | **100.0±0.0** | **99.9±0.1** | **96.1±0.1** | **94.0±0.2** | **92.2±0.3** | **91.7±0.2** | **93.6±0.3** |
| | PGD+FSO $L_{2\_\infty}^{2X}, \frac{1}{2}\varepsilon_{2\_\infty}$ | 88.7±0.9 | 100.0±0.1 | 97.8±1.2 | 78.9±1.1 | 73.5±0.6 | 68.7±0.4 | 64.7±0.5 | 69.5±0.5 |
| | PGD+FSO $L_{2\_\infty}^{3X}, \frac{1}{3}\varepsilon_{2\_\infty}$ | 77.5±0.7 | 100±0.2 | 93.0±0.1 | 61.6±0.3 | 59.1±0.6 | 58.7±0.1 | 50.2±0.1 | 57.1±0.3 |
| | PGD+FSO $L_{2\_\infty}^{3X}, \frac{2}{3}\varepsilon_{2\_\infty}$ | 94.2±0.5 | 100±0.1 | 98.8±0.1 | 88.4±0.2 | 83.8±0.1 | 80.2±0.3 | 78.6±0.4 | 83.1±0.1 |

(thus after 30 steps, the attack strength is still smaller then the control value). For other attack method, the improvement of FSO becomes less but FSO still maintains the best performance.

Ensemble models are useful to enhance the transferability in general. In Table 2, we show the results obtained by an ensemble model (RN-34+RN-152+DN-121). The transferability of multi-step PGD attack with $L_2$ and $L_\infty$ norms improves a lot by this ensemble model, partly due to the stabler perturbation direction which leads to faster growth of perturbation strength. Again, the transferability is still significantly improved by FSO under the $L_{2\_\infty}$ norm. The transferability is even better for FSO under the $L_2$ norm, because there is no constraint on the $L_\infty$ norm of perturbations.

By using the $L_{2\_\infty}$ norm, the $L_\infty$ norm is allowed to be $m$ times $\varepsilon_\infty$. We further conducted the experiment for FSO under smaller perturbation strengths: $\frac{1}{2}\varepsilon_{2\_\infty}$ for $m = 2$ and $\frac{1}{3}\varepsilon_{2\_\infty}$ for $m = 3$. In these cases, the upper limit of $L_\infty$ norm for the perturbation is also $\varepsilon_\infty$ (but the upper limit of $L_2$ norm is only half or one third $\varepsilon_2$). The results are also shown in Table 2. For the case $\frac{1}{2}\varepsilon_{2\_\infty}$ and $m = 2$, We can see that FSO still achieves better results compared to PGD with $L_\infty$ norm; For the case $\frac{1}{3}\varepsilon_{2\_\infty}$ and $m = 3$, the performance obtained with FSO becomes inferior because the limit of $L_2$ norm is too small.

Other attack methods such as MI, VR, SGM, IR, and TI, can also be naturally incorporated in FSO. For these attack methods, we also observed significant enhancement of transferability by using FSO under the $L_{2\_\infty}$ norm.

## 5.4 COMPARISON OF PERTURBED IMAGES

In Figure 5, we show the comparison of perturbed images using different attack methods. The six columns of images are obtained by PGD, SGM, VR, TI, MI, and IR attack methods to obtain $\boldsymbol{g}^t$, respectively; whereas the five rows are obtained by traditional multi-step attack method (with $L_2$ and

$L_\infty$ norms) and FSO under $L_{2\_\infty}$ norm with (1) $m = 2$ and $\varepsilon = \frac{1}{2}\varepsilon_{2\_\infty}$; (2) $m = 3$ and $\varepsilon = \varepsilon_{2\_\infty}$, and FSO under $L_2$ norm, respectively. In general, we can see that perturbations obtained with FSO under $L_{2\_\infty}$ norm are more unconspicuous. In particular, perturbations in the second row are much more imperceptible because the small attack strength, while it maintains comparable or even better transferability than perturbations obtained with multi-step attack under $L_\infty$ norm.

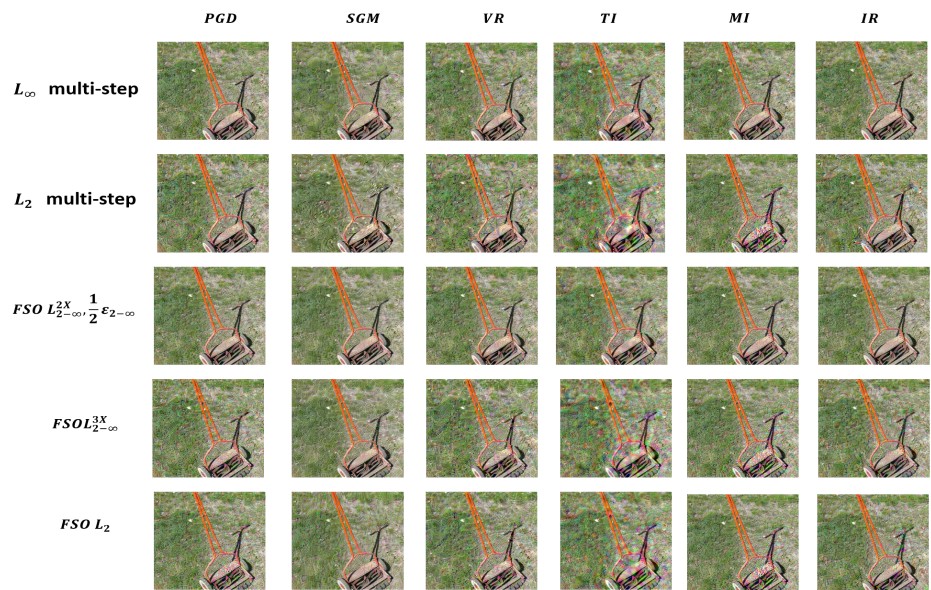

Figure 5: Visualization of adversarial examples generated by various methods and their combination with FSO.

## 6 CONCLUSION

We developed a Fixed Strength Optimization (FSO) method to generate adversarial examples. In FSO, the optimization of adversarial examples are directly performed under the constraint of fixed perturbation strength, where the strength is defined as the norm of the perturbation. FSO can significantly improve both the convergence speed and transferability. The perturbation is well optimized in a few ($\leq 10$) steps. We also propose a combined norm, the $L_{2\_\infty}$ norm, for adversarial examples to balance the attack on semantic information and introduction of noise. By incorporating the combined norm into FSO, our numerical experiments show improved attack transferability and high imperceptibility of perturbations.

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
