# OpenReview forum: "Fixed Strength Optimization Enhances Adversarial Attacks"
_ICLR.cc/2025/Conference — ICLR 2025 Conference Withdrawn Submission_

### Official Review · Reviewer_fk7p · 2024-10-29

**Soundness:** 2
**Presentation:** 2
**Contribution:** 2
**Rating:** 5
**Confidence:** 4

**Summary:**

This paper proposes Fixed Strength Optimization (FSO) to accelerate adversarial example convergence and enhance their transferability.

**Strengths:**

1 L_{2_\infty} is novel and the motivation is interesting.

**Weaknesses:**

1 How can the proposed method to be applied to advanced transfer attack algorithms? How are the experimental results?

2 Why are the results for the white-box scenario not shown in Table 1?

3 Figure 5 does not show the original image, making it impossible to compare the imperceptibility of noise added by different methods.

**Questions:**

See Weaknesses.

---

### Official Review · Reviewer_nMnF · 2024-10-31

**Soundness:** 1
**Presentation:** 1
**Contribution:** 2
**Rating:** 1
**Confidence:** 3

**Summary:**

This work proposes an optimization technique for adversarial attacks that accelerates convergence of adversarial evasion attacks by updating the perturbation by walking on a mixture of the L2 and L-infinity norm.

**Strengths:**

+ attempts to improve optimization of transfer-based adversarial attacks

**Weaknesses:**

# Weaknesses:

- the paper is difficult to read even for experts
- the assumptions should be clarified
- the evaluation is not convincing

-------------------------------

# Comments:

**Clarity of the paper**

The paper is really difficult to read (also for experts) and there are multiple sentences that are unclear. Below a (probably non-exhaustive) list of sentences that are obscure:

- Given an attack method, the actual perturbation strength (perturbation radius) gradually increases during the multi-step process until it reaches the preset upper limit, or never reaches the upper limit until the end
- The optimization in FSO is difficult if the L∞ norm is used, because the tangential component is always zero;
- A perturbation obtained with a multi-step L∞ attack method is not an optimized perturbation based on L∞ norm.
- selectively modifies the largest components to enforce sparsity
- thereby boosting the transferability
- suppressing the attack of semantic information
- These pixels with unstable sign of gt
- Good balance of perturbation

**Concerns regarding the soundness**

- It's not clear if the authors enforce the constraints of the image representation, i.e., pixels in [0, 1], quantized in 256 levels.
- It's not clear why the authors introduce and explain all the methods in sect. 2.1
- "the tangential component (to the ε-sphere) of the perturbation direction gt is usually small compared to the normal component, which lead to slow convergence of the multi-step method. " This should be supported by some experimental evidence (e.g., measuring the tangential component).

**Concerns regarding the evaluation.**
- The method is not tested with robust models.
- the hyperparameters are not reported, and the authors claim that PGD has slow convergence but they don't seem to consider different choices of step sizes and other optimization techniques

Overall, the main issue of this paper is that it is difficult to understand what the authors are doing. The manuscript should be enhanced substantially to make it clearer and demonstrate support for the claims made by the authors.

**Questions:**

The paper mainly needs to be rewritten. There are too many sentences that are unclear and make the content of the work really difficult to evaluate.

Additional questions:
- is the same subset of 1000 samples used throughout the experiments?
-

---

### Official Review · Reviewer_oCXA · 2024-11-02

**Soundness:** 2
**Presentation:** 2
**Contribution:** 2
**Rating:** 3
**Confidence:** 5

**Summary:**

The authors present an innovative gradient-based adversarial attack method aimed at improving the efficiency of such attacks. The paper introduces a novel approach that employs a fixed-strength attack in conjunction with a combined norm to execute adversarial attacks. The experimental results presented in the paper demonstrate the potential effectiveness of the proposed method. While the paper's technical contribution is promising, it would benefit from additional refinement and clarification in certain areas to fully realize its contribution to the field.

**Strengths:**

1. The paper provides a valuable exploration into the role of the $\epsilon$-sphere in enhancing the transferability of adversarial examples, which is a significant aspect of adversarial machine learning. This should provide insight for researchers and practitioners in the field.

2. The introduction of a combined norm to facilitate the search for adversarial examples is an innovative concept that has shown to be beneficial according to the results presented.

**Weaknesses:**

1. The title could be refined to more accurately reflect the scope of the paper, which primarily focuses on the transferability of adversarial examples. Additional empirical data on the efficiency of the attack, such as the number of steps or computation time required to achieve a certain success rate, would strengthen the paper's claims on the efficiency of the attack (rather than only the transferability).

2. The background section, while comprehensive, could be condensed to maintain focus and relevance to the current work. Current version of the paper is somewhat lengthy and not helpful for the readers to gain a deeper understanding of this paper's method.

3. In Section 3.1, the assertion regarding the tangential component's magnitude could be substantiated with preliminary experiments or additional evidence to enhance its credibility. Currently, it is not clear why the magnitude is small. Moreover, the claim under $L_\infty$ norm that it is always zero is not clearly explained either.

4. Section 4 would benefit from a more detailed explanation of the methodology behind Figure 2(a), including the meaning of the notations 'single proj.' and 'double proj.'

5. The organization of the sections could be optimized for readability. Section 4, for instance, might be more appropriately presented as a subsection within the methodology section (Section 3).

6. Clarification is needed regarding the shaded area in Figure 3, as it is not currently visible.

7. The experimental settings described in Section 5 require more comprehensive documentation to enable reproducibility.

8. The introduction of PGD FSO in Section 5.3 should be accompanied by a thorough explanation of the method's construction.

9. The visualization presented in Section 5.4 could be relocated to an appendix unless it provides critical insights that are essential to the main text.

10. Minor corrections:
    (1) In Section 4, the phrase "adversarial examples ... is not optimized" should be corrected to "adversarial examples ... are not optimized".
    (2) The label "Pertubation Strength" in Figure 2(b) should be corrected to "Perturbation Strength" to address the typographical error.

**Questions:**

1. Could the authors provide further details on the rationale behind the small or zero tangential component? Is this based on empirical observations or preliminary experiments?

2. Additional information on the creation of Figure 2(a), including the data and methods used, would be beneficial for understanding the results presented.

3. Is there an issue with the depiction of Figure 3, specifically regarding the shaded area that is not currently visible?

4. A more detailed explanation of how the PGD FSO method is constructed would be appreciated for a clearer understanding of its implementation.

---

### Official Review · Reviewer_Sr8J · 2024-11-04

**Soundness:** 3
**Presentation:** 2
**Contribution:** 2
**Rating:** 5
**Confidence:** 4

**Summary:**

This paper proposes a method called Fixed Strength Optimization (FSO) for accelerating the convergence of generated adversarial samples under a preset fixed attack strength. In addition, the paper introduces a new paradigm ($L_{2- ∞}$ paradigm) that combines the $L_2$ paradigm and the $L_∞$ paradigm, which is used to regulate the attack direction during the optimization process in order to balance the attack strength of the semantic information and noise components. Through numerical experiments, the authors demonstrate that the FSO method can achieve faster convergence and higher transfer capability with the same attack strength, and the use of the combined paradigm improves the transferability of the adversarial examples.

**Strengths:**

1.The FSO method avoids the problem of gradual increase of attack intensity in traditional multi-step methods by fixing the optimization framework of attack intensity.
2.The paper demonstrates in detail that the advantages of the FSO method in several aspects, such as convergence speed and transferability, by comparing it with several existing methods; the experimental results support the validity of the approach.

**Weaknesses:**

1. The paper did not state whether all experiments were non-targeted or targeted attacks and the selected methods for comparison are too old.
2. A similar approach was used in the previous work DAoB[1], where the adversarial perturbation was first pulled to the constraint boundaries and then the search for adversarial examples was continued. The paper needs to point out the differences between the approaches and analyze them experimentally.
3. In Table 2, the performance of PGD+FSO+$L_{2-∞}^{3X}$ is not better than PGD+$L_2$. It is better to add experiments for PGD+FSO+$L_2$ in Table 1 to demonstrate the advantages of the proposed $L_{2-∞}$.
4. For the quality of adversarial examples, objective metrics should be given in addition to visualization, such as SSIM.
5. Theoretical proof of the proposed method is necessary.

[1] DAoB: A Transferable Adversarial Attack via Boundary Information for Speaker Recognition Systems

**Questions:**

1. What is the meaning of $L_{2-∞}^{3X}$ and $L_{2-∞}^{2X}$ in the paper?
2. The presentation of tangential component of $g_t$ is unclear, and how to solve the tangential component problem by $L_{2-∞}$?
3. What is the difference of the projection operator between PGD and FSO?
4. The key to improve the transferability seems to lie in the value of m. As m > 1, the corresponding  $L_∞$ is also getting larger, which is not a fair comparison with other methods.

---

### Note · Authors · 2024-11-25

I have read and agree with the venue's withdrawal policy on behalf of myself and my co-authors.